# Diagnostic Accuracy of the Deep Learning Model for the Detection of ST Elevation Myocardial Infarction on Electrocardiogram

**DOI:** 10.3390/jpm12030336

**Published:** 2022-02-23

**Authors:** Hyun Young Choi, Wonhee Kim, Gu Hyun Kang, Yong Soo Jang, Yoonje Lee, Jae Guk Kim, Namho Lee, Dong Geum Shin, Woong Bae, Youngjae Song

**Affiliations:** 1Department of Emergency Medicine, College of Medicine, Hallym University, Chuncheon 24252, Korea; chy6049@naver.com (H.Y.C.); drkang9@gmail.com (G.H.K.); amicoys@gmail.com (Y.S.J.); yong0831@naver.com (Y.L.); gallion00@gmail.com (J.G.K.); 2Division of Cardiology, Department of Internal Medicine, College of Medicine, Hallym University, Chuncheon 24252, Korea; namholee@hallym.or.kr (N.L.); blaudg@naver.com (D.G.S.); 3VUNO Inc., Seoul 06541, Korea; iorism@vuno.co (W.B.); yjsong@vuno.co (Y.S.)

**Keywords:** deep learning, ST elevation myocardial infarction, electrocardiography, predictive value of tests

## Abstract

We aimed to measure the diagnostic accuracy of the deep learning model (DLM) for ST-elevation myocardial infarction (STEMI) on a 12-lead electrocardiogram (ECG) according to culprit artery sorts. From January 2017 to December 2019, we recruited patients with STEMI who received more than one stent insertion for culprit artery occlusion. The DLM was trained with STEMI and normal sinus rhythm ECG for external validation. The primary outcome was the diagnostic accuracy of DLM for STEMI according to the three different culprit arteries. The outcomes were measured using the area under the receiver operating characteristic curve (AUROC), sensitivity (SEN), and specificity (SPE) using the Youden index. A total of 60,157 ECGs were obtained. These included 117 STEMI-ECGs and 60,040 normal sinus rhythm ECGs. When using DLM, the AUROC for overall STEMI was 0.998 (0.996–0.999) with SEN 97.4% (95.7–100) and SPE 99.2% (98.1–99.4). There were no significant differences in diagnostic accuracy within the three culprit arteries. The baseline wanders in false positive cases (83.7%, 345/412) significantly interfered with the accurate interpretation of ST elevation on an ECG. DLM showed high diagnostic accuracy for STEMI detection, regardless of the type of culprit artery. The baseline wanders of the ECGs could affect the misinterpretation of DLM.

## 1. Introduction

Although the mortality of acute myocardial infarction (AMI) has been improving recently with coronary reperfusion therapy, AMI is still the leading cause of death worldwide [1,2]. AMI can be divided into ST elevation myocardial infarction (STEMI) and non-ST elevation myocardial infarction (NSTEMI). STEMI has more severe complaints than NSTEMI, and a higher mortality rate with rapid disease progression [3,4].

Since STEMI requires urgent reperfusion therapy, the quick and accurate interpretation of the electrocardiogram (ECG) is essential. The interpretation of STEMI on ECG is a difficult task for emergency physicians and cardiologists [2].

ECG machines have their own automatic machine interpretation programs according to the manufacturers. Clinicians have used the automatic ECG interpretation provided by ECG machines to determine whether the ECG is a real STEMI. However, previous studies reported that the inaccuracy of the automatic interpretation of ECG machines ranged from 5.9% to as high as 29%. Thus, the automatic ECG interpretation of STEMI remains unreliable [5,6,7]. Although several studies have been conducted to improve the accuracy of ECG machine interpretation, obtaining an accurate ECG interpretation remains difficult [8,9].

Deep learning has recently been applied to improve the accuracy of ECG analysis. Additionally, efforts to improve the accuracy of STEMI-ECG using deep learning have been actively attempted [1,10,11,12]. However, the accuracy of STEMI-ECG interpretation by deep learning has not yet reached a clinically applicable level to diagnose real STEMI.

This study aimed to demonstrate that the application of deep learning can improve the accuracy of STEMI-ECG interpretation.

## 2. Materials and Methods

### 2.1. Study Design

This was a single center, retrospective, observational study. The recruitment period was from January 2017 to December 2019. This study was approved by the Hallym University Kangnam Sacred Heart Hospital Institution Review Board in November 2019 (No. HKS 2019-10-021-001).

By searching electronic medical records during the recruitment period, we recruited patients with AMI who received more than one stent insertion for culprit artery occlusion after visiting the emergency room in the Hallym University Kangnam Sacred Heart Hospital. The culprit coronary artery was defined as any vessel with acute thrombotic total or subtotal occlusion [13].

A single ECG machine (MAC 5500 HD, GE Healthcare, Chicago, IL, USA) was used in the emergency room during the recruitment period for the STEMI group. Two trained cardiologists with sufficient experience interpreted and confirmed STEMI-ECGs among AMI patients using reliable STEMI diagnostic criteria [14]. The cardiologists were board certified physicians in cardiology and had worked in a university hospital for more than ten years.

The STEMI group was categorized into three groups according to the type of culprit artery receiving coronary intervention. The culprit arteries were the left anterior descending artery (LAD), left circumflex artery (LCX), and right coronary artery (RCA). We collected information on baseline characteristics and analyzed the success rate of STEMI detection using an ECG machine.

Patients with an ECG interpretation of “normal sinus rhythm (NSR) and normal ECG” were considered as the NSR group. The ECGs of the NSR group were collected using the Cardiology Information System (MUSE^®^, GE Healthcare, Chicago, IL, USA) during the same recruitment period as patients with STEMI.

### 2.2. Outcomes

The primary outcome was the diagnostic accuracy of the deep learning model (DLM) for STEMI according to the culprit arteries. The outcomes were measured using area under the receiver operating characteristic curve (AUROC), sensitivity (SEN), specificity (SPE), positive predictive value (PPV), and negative predictive value (NPV) using the Youden index.

### 2.3. Application of Deep Learning Model

We externally validated the ECG dataset to measure the STEMI detection performance of the DLM between STEMI-ECGs and NSR-ECGs. DLM was also applied to measure the difference in STEMI detection performance according to culprit artery occlusion.

The data for training and internal validation of DLM were the ECGs of patients who visited the Cardiology and Heart Surgery Specialty Center in the Sejong General Hospital from October 2016 to March 2019. DLM learned and trained with 122,813 ECGs from 55,451 patients. After training, DLM was internally validated with 26,018 ECGs from 11,883 patients, which were randomly separated from the ECGs for training. In the internal validation for STEMI detection, DLM resulted in 0.983 AUROC (95% confidence interval (CI) 0.976–0.989), SEN 85.0% (95% CI 78.1–91.6), and SPE 96.0% (95% CI 95.8–96.2).

The flow of the DLM is represented in Figure 1 in the development of the model architecture. Twelve-lead raw ECG with differentiated data and the P, QRS, and T section information were concatenated as the input. Ten-second ECG was split to obtain four ensembled outputs for 2.5 s each, to ensure stable performance. The weight shared encoder, which is based on the ResNet structure with some modifications (Appendix A), outputs each lead information [15]. These were concatenated and used as the input of the classifier, fully connected layers, and a sigmoid layer. The classifier has a probability between STEMI-ECGs and NSR-ECGs.

### 2.4. Statistical Analysis

The data were compiled using a standard spreadsheet application (Excel, Microsoft, Redmond, WA, USA) and analyzed using the Statistical Package for the Social Sciences (SPSS) 26.0 KO for Windows (SPSS Inc., Chicago, IL, USA). We generated descriptive statistics and presented them as frequencies and percentages for categorical variables. Continuous variables are presented as mean with standard deviation (mean ± SD) for parametric data or median with interquartile range for nonparametric data. Normality for continuous variables was tested using the Shapiro–Wilk test. To identify the correlation between factors and autointerpretation of ECG, the chi-square test or Fisher’s exact test was used for categorical variables. An independent t-test (parametric data) or Mann–Whitney test (nonparametric data) was used for continuous variables. The external validation performance of the DLM was evaluated using the AUROC, SEN, and SPE using a two sided 95% CI. To compare AUROC among the three different culprit artery groups, the significance was statistically tested using DeLong’s test. Statistical significance was set at *p* < 0.05.

## 3. Results

### 3.1. Diagnostic Accuracy of DLM between STEMI-ECGs versus NSR-ECGs

A total of 60,157 ECGs were obtained. These included 117 STEMI-ECGs and 60,040 NSR-ECGs (Figure 2).

The baseline characteristics of the experimental groups are shown in Table 1. There were no significant factors affecting the success of STEMI detection by automated ECG interpretation.

DLM showed an AUROC of 0.998 (95% CI 0.996–0.999) for STEMI detection compared with NSR-ECGs in the external validation. Based on the Youden index score, SEN = 0.974 and SPE = 0.992 (Figure 3). In the analysis by the sort of culprit arteries, the AUROC performance of STEMI detection was as follows: RCA, 0.998 (CI 0.995–0.999); LAD, 0.998 (CI 0.996–0.999); LCX, 0.999 (CI 0.998–1.000). There were no significant differences in the comparison of AUROC performance according to culprit arteries (DeLong’s test; RCA vs. LAD, *p* = 0.398; LAD vs. LCX, *p* = 0.369; RCA vs. LCX, *p* = 0.169).

The DLM also showed a high SEN and SPE, as shown in Table 2. In the detection of the overall STEMI-ECGs, the SEN and SPE were 97.4% and 99.2%, respectively. In the analyses by culprit arteries, all SEN and SPE were in the range of 95–100%. It demonstrated a very high overall NPV (99.9%), with the NPV above 99% regardless of the type of culprit artery. PPV, on the other hand, had an extremely low value (Total: 20.2%; RCA: 4.6%; LAD: 15.7%; LCX: 3.2%).

### 3.2. Analysis for False Positive and False Negative Results

We found that the DLM detected 412 NSR-ECGs as STEMI-ECGs, which suggested false positive cases in Table 3. These false positive ECGs included 345 ECGs with a baseline wander (83.7%, 345/412). Ten real STEMI patients were identified in the chart review by cardiologists. Nevertheless, there was no significant difference in real STEMI-ECGs regardless of the existence of a baseline wander (*p* = 0.49). In the measurement of the effect of a baseline wander, the results suggested that the interpretation of ST elevation was significantly impeded by a baseline wander (*p* = 0.001). The six examples of false positive cases are shown in Appendix A using 12-lead ECGs and heatmap images.

Three STEMI-ECGs were false negative by DLM. Two ECGs were the inferior wall (all RCA culprit arteries) and an anteroseptal wall STEMI (LAD culprit artery). All false negative ECG results showed baseline wanders.

## 4. Discussion

This study demonstrated that DLM could significantly improve the accuracy of STEMI-ECG interpretation. DLM showed high diagnostic performance for STEMI detection (AUROC, 0.998; SEN, 97.4%; SPE, 99.2%). In the analysis according to three different culprit coronary arteries, the diagnostic accuracy of LAD, LCX, and RCA STEMI was greater than AUROC 0.99. In addition, the sensitivity and specificity of DLM were >95%. The PPV was only 20.2% while having a high NPV of more than 99%. We believe that the low PPV is linked to the cohort’s low STEMI prevalence (117 STEMI-ECG vs. 60,040 NSR-ECG; prevalence = 0.2%). We also found that the baseline wander of ECGs could affect the misinterpretation of DLM by analyzing false positive and false negative ECGs.

We applied the DLM to classify STEMI-ECGs from NSR-ECGs. Compared with typical convolutional neural network (CNN) models, which have been widely used in ECG studies, DLM has some advantages for STEMI detection [16,17,18]. First, the DLM processes the 12-lead (each channel) information separately by applying weight shared encoders for each channel, which can reduce information loss by including each lead. Typical models, which have a single encoder for 12 leads, usually aggregate channel information in the first layer. However, since the ST elevation feature could appear on a few leads associated with the obstruction of culprit arteries, the DLM showed better performance in the STEMI detection task than the single encoder. Second, the DLM can optimally process the ECG feature input. The major ECG feature input is as follows: P wave, QRS complex, T wave, and the shape, amplitude, and latency of each wave. In the application of a conventional CNN with a single encoder, the aggregation of this channel information can impede rapid ECG input processing.

In the analysis of false positive and false negative ECGs, we found that a baseline wander significantly affected the accuracy of DLM. A baseline wander was detected in 83.7% (345/412) in false positive ECGs and 100% (3/3) in false negative ECGs. In the selection of STEMI-ECGs by cardiologists, we removed those with a baseline wander for correct interpretation. However, when extracting NSR-ECGs through a cardiology information system, several NSR-ECGs with baseline wander were included, as the filtering function for baseline wander was not embedded in this system. The baseline wander might have originated from unsolved human factors such as movement, shivering, or poor contact of ECG leads with dry or hairy skin [19]. Considering these results, we anticipate that the application of the wander filtering system will improve the diagnostic accuracy of DLM.

The analysis of false positive cases also showed that the control group included 10 STEMI (3 ECGs without baseline wander and 7 ECGs with baseline wander). This result was caused by setting the control group using the autointerpretation of ECG machines. In addition to the 10 STEMI ECGs, true negative results included 402 NSR-ECGs. Unfortunately, we could not identify the cause of false positive results because the DLM did not report the analysis of misinterpretation. Additionally, the ST elevation of ECGs without a baseline wander was significantly higher than that of ECGs with a baseline wander. This result also demonstrated that a baseline wander could significantly interfere with the interpretation of ST elevation by DLM [20].

DLM allows the possibility of STEMI to be detected with high accuracy in a short period in circumstances when it is difficult to determine the STEMI on the ECG in the prehospital stage, or for medical professionals who are unfamiliar with the ECG reading at the primary medical institution. This might help STEMI patients by allowing them to go to a cardiovascular center where revascularization can be performed quickly.

This study has some limitations. First, the sample size of the STEMI group (*n* = 117) was smaller than that of the NSR group (*n* = 60,040) during the same recruitment period. Although the model showed 100% accuracy in the autointerpretation of LCX-STEMI, the sample size of LCX-STEMI was smaller than that of LAD-STEMI and RCA-STEMI. Therefore, the accuracy of the autointerpretation of ECG machines or DLM might differ in the study of a larger sample size of STEMI. Second, the control group was selected using the autointerpretation of ECG machines, which resulted in the misinterpretation of 10 cases with STEMI. Third, this was a retrospective single center study, which limits its generalizability. Fourth, only one type of ECG machine was used in this study. If the algorithm for the autointerpretation of EGCs varies according to the type or manufacturer of ECG machine, the accuracy of autointerpretation might change. Fifth, some ECGs with left ventricular hypertrophy (LVH), left bundle branch block (LBBB), and right bundle branch block (RBBB) were not included in the STEMI group. ECGs such as LVH and LBBB, according to the myocardial infarction definition, can simulate ST deviation [14]. For STEMI-ECG with LBBB, the Sgarbossa criteria is known. Although the RBBB ECG is not an exclusion criteria in the definition of myocardial infarction, we took into account the difficulties of interpreting ST elevation on precordial leads such as V1-2 in a clinical setting. As a result, during the external validation process, we eliminated the STEMI-ECG with LVH or LBBB. The diagnostic accuracy of the deep learning model will be affected if these ECGs are incorporated. Sixth, we used C statistics to assess the diagnostic accuracy of DLM. Although C statistics have been routinely utilized to assess the predictive power of models, their correlation to clinical outcomes has been questioned. Novel statistical indices such as “net benefit” have been proposed for measuring the diagnostic performance of tools or the prediction power of models to counteract this flow [21]. To compute “net benefit”, we attempted to establish a “harm to benefit ratio”. We were unable to discover the publication claiming a “harm to benefit ratio” for myocardial infarction despite a comprehensive search. It is recommended that further study be conducted before using a robust prediction model such as “net benefit”.

## 5. Conclusions

DLM showed high diagnostic accuracy for STEMI detection regardless of the type of culprit artery. Baseline wanders of the ECGs could affect the misinterpretation of DLM.

## Figures and Tables

**Figure 1 jpm-12-00336-f001:**
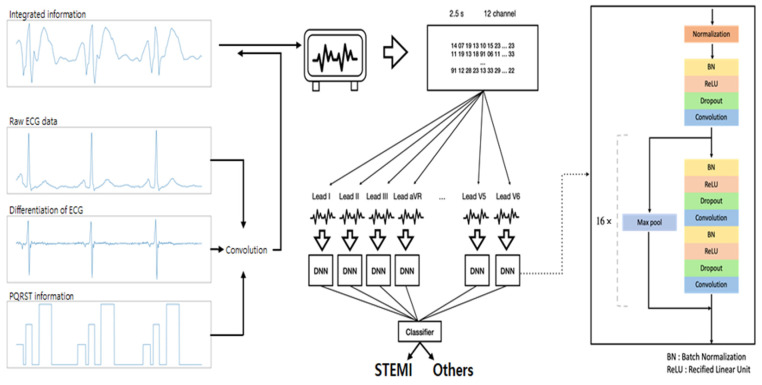
Model architecture of deep learning model. ECG, electrocardiogram; STEMI, ST elevation myocardial infarction; DNN, deep neural networks.

**Figure 2 jpm-12-00336-f002:**
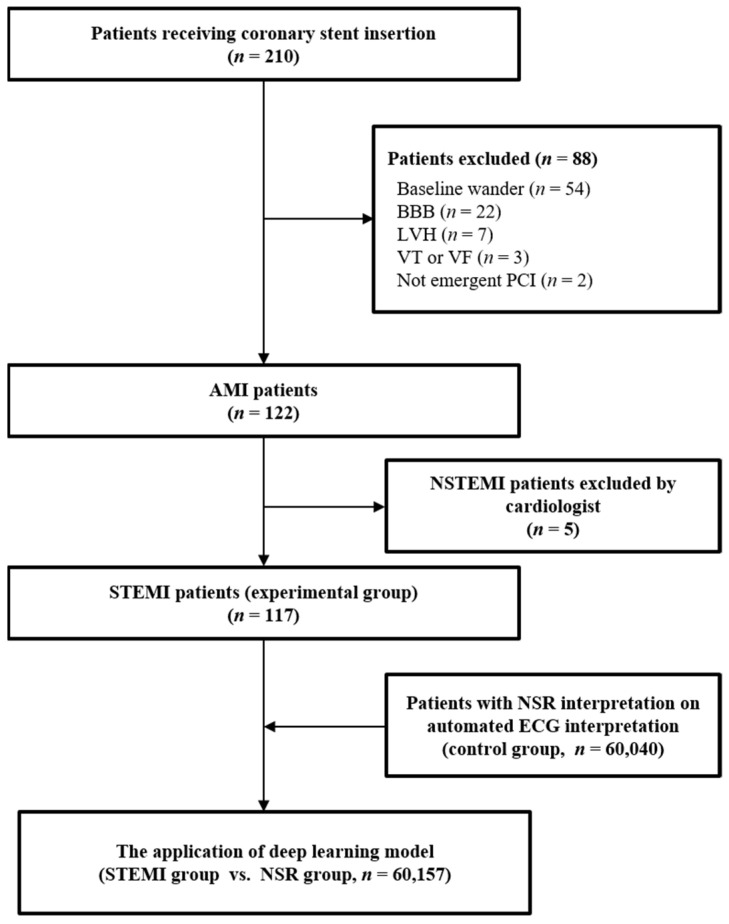
Flow diagram for this study. The patients having the ECG interpretation of “normal sinus rhythm and normal ECG” were included in the NSR group. BBB, bundle branch block; LVH, left ventricular hypertrophy; VT, ventricular tachycardia; VF, ventricular fibrillation; PCI, percutaneous coronary intervention; AMI, acute myocardial infarction; NSTEMI, non-ST elevation myocardial infarction; STEMI, ST elevation myocardial infarction; NSR, normal sinus rhythm; ECG, electrocardiogram.

**Figure 3 jpm-12-00336-f003:**
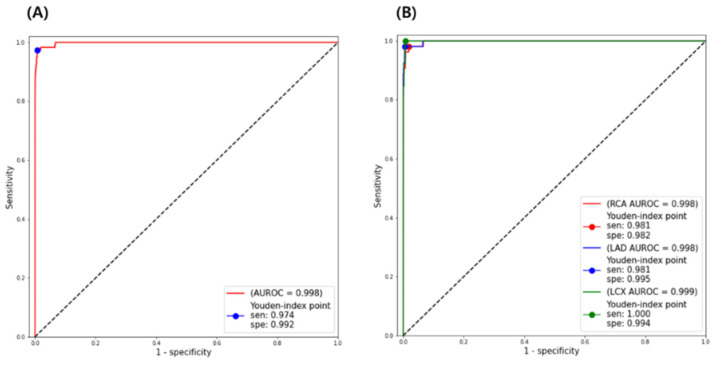
Diagnostic accuracy of the deep learning model for ST elevation myocardial infarction detection compared with normal sinus rhythm. (**A**) Overall accuracy and (**B**) the comparative accuracy according to culprit coronary arteries. STEMI, ST elevation myocardial infarction; ROC, receiver operating characteristic; AUROC, area under the receiver operating characteristic; SEN, sensitivity; SPE, specificity; NSR, normal sinus rhythm; RCA, right coronary artery; LAD, left anterior descending artery; LCX, left circumflex artery.

**Table 1 jpm-12-00336-t001:** Baseline characteristics of the experimental group.

	Experimental Group Diagnosed with STEMI (*n* = 117) STEMI Detection by the Automated ECG Interpretation		
Factors	Failure (*n* = 31)	Success (*n* = 86)	*p*-value *
Sex, male	24 (77.4%)	70 (81.4%)	0.63
Age, years	61.3 ± 10.5	58.2 ± 11.8	0.41
Underlying diseases			
DM	6 (19.4%)	15 (17.4%)	0.81
HTN	16 (51.6%)	37 (43%)	0.41
Angina	0 (0%)	1 (1.2%)	1.0
CHF	0 (0%)	0 (0%)	NA
CKD	1 (3.2%)	3 (3.5%)	1.0
Past history			
Smoking, pack year	20.3 ± 18.1	19.5 ± 17.6	0.92
Previous PCI	0 (0%)	1 (1.2%)	1.0
PO medication			
Aspirin	1 (3.2%)	4 (4.7%)	1.0
Antiplatelet	0 (0%)	2 (2.3%)	1.0
ACE inhibitor	0 (0%)	0 (0%)	NA
Statin	0 (0%)	1 (1.2%)	1.0
Laboratory findings			
Troponin I, pg/mL	21.6 (0.2–102.6)	3.2 (0–128.5)	0.29
CK-MB, ng/mL	3.4 (1.7–9.9)	2.7 (1.3–7.2)	0.34
BNP, pg/mL	41 (10.3–167.2)	25.9 (10.3–65.5)	0.31
Cr, mg/dL	0.9 (0.7–1)	0.9 (0.7–1)	0.93
Vital signs			
HR, bpm	73.7 ± 17.5	77.3 ± 19.6	0.29
SBP, mmHg	132.1 ± 25.8	133.7 ± 28.3	0.57
DBP, mmHg	84 ± 18.5	83.6 ± 17.7	0.50
Patient outcomes			
Arrest	1 (3.2%)	9 (10.5%)	0.28
ECMO	1 (3.2%)	5 (5.8%)	1.0
TTM	0 (0%)	0 (0%)	NA
Pacemaker	1 (3.2%)	3 (3.5%)	1.0
MV	2 (6.5%)	9 (10.5%)	0.72
Hospital admission, day	5.4 ± 3.9	5.9 ± 4.2	0.42
ICU stay, day	4.3 ± 4.4	3.8 ± 2.5	0.44
Survival	30 (96.8%)	82 (95.3%)	1.0
Culprit artery			0.132
LAD	14 (45.2%)	27 (31.4%)	
LCX	0 (0%)	7 (8.1%)	
RCA	13 (41.9%)	25 (29.1%)	
LAD-LCX	2 (6.5%)	4 (4.7%)	
LAD-RCA	1 (3.2%)	12 (14%)	
LCX-RCA	0 (0%)	6 (7%)	
LAD-LCX-RCA	1 (3.2%)	5 (5.8%)	

* Calculated using the chi-square test or Fisher’s exact test for categorical data. All continuous variables were parametric except for laboratory findings. Nonparametric data in the laboratory findings were tested using the Mann–Whitney test. Statistical significance was set at *p* < 0.05. Abbreviations: ECG, electrocardiogram; STEMI, ST elevation myocardial infarction; DM, diabetes mellitus; HTN, hypertension; CHF, congestive heart failure; CKD, chronic kidney disease; PO, per oral; PCI, percutaneous coronary intervention; ACE, angiotensin-converting enzyme; CK-MB, creatine kinase MB; BNP, brain natriuretic peptide; Cr, creatine; HR, heart rate; SBP, systolic blood pressure; DBP, diastolic blood pressure; ECMO, extracorporeal membrane oxygenation; TTM, target temperature management; MV, mechanical ventilation; ICU, intensive care unit; bpm, beats per minute; NA, not applicable; LAD, left anterior descending artery; LCX, left circumflex artery; RCA, right coronary artery.

**Table 2 jpm-12-00336-t002:** Diagnostic accuracy of deep learning model for ST elevation myocardial infarction detection comparing with normal sinus rhythm.

Coronary Culprit Artery	AUROC (95% CI)	Sensitivity (%) (95% CI)	Specificity (%) (95% CI)	Positive Predictive Value % (95% CI)	Negative Predictive Value % (95% CI)
Overall STEMI (*n* = 117)	0.998 (0.996–0.999)	97.4 (95.7–100)	99.2 (98.1–99.4)	20.2 (9.2–21.9)	99.9 (99.9–100.0)
RCA-STEMI (*n* = 63; 38 RCA, 13 LAD-RCA, 6 LCX-RCA, 6 LAD-LCX-RCA)	0.998 (0.995–0.999)	98.1 (95.8–100)	98.2 (93.5–99.4)	4.6 (1.3–11.7)	99.9 (99.9–100.0)
LAD-STEMI (*n* = 66; 41 LAD, 6 LAD-LCX, 13 LAD-RCA, 6 LAD-LCX-RCA)	0.998 (0.996–0.999)	98.1 (95.4–100)	99.5 (99.4–99.8)	15.7 (15.7–35.2)	99.9 (99.9–100.0)
LCX-STEMI (*n* = 25; 7 LCX, 6 LAD-LCX, 6 LCX-RCA, 6 LAD-LCX-RCA)	0.999 (0.998–1.000)	100 (100–100)	99.4 (99.3–99.9)	3.2 (3.2–66.6)	100.0 (99.9–100.0)

Abbreviations: STEMI, ST elevation myocardial infarction; AUROC, area under the receiver operating characteristic; CI, confidence interval; RCA, right coronary artery; LAD, left anterior descending artery; LCX, left circumflex artery.

**Table 3 jpm-12-00336-t003:** Analysis of false positive cases to measure the effect of baseline wander on the accuracy of deep learning model.

Control Group (*n* = 412)	Baseline Wander (−) (*n* = 67)	Baseline Wander (+) (*n* = 345)	*p*-Value *
Real STEMI ^†^	3 (4.5%)	7 (2%)	0.49
Not STEMI	64 (95.5%)	338 (98%)	
STE ^‡^	15 (22.4%)	29 (8.4%)	0.001
No STE	52 (77.6%)	316 (91.6%)	

* Calculated using the chi-square test or Fisher’s exact test for categorical data. Statistical significance was set at *p* < 0.05. ^†^ Compatible with the inclusion criteria for STEMI in the experimental group. ^‡^ Compatible with the definition criteria for STEMI on ECG. Abbreviations: STEMI, ST elevation myocardial infarction; STE, ST elevation.

## Data Availability

The datasets generated during the current study are available from the corresponding author upon reasonable request.

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
