# Peer review of "Diagnostic Accuracy of the Deep Learning Model for the Detection of ST Elevation Myocardial Infarction on Electrocardiogram"

_jpm, 2022, doi:10.3390/jpm12030336_

Round 1

Reviewer 1 Report

Deep Learning of ECG tracing for ST elevation (machine learning) does not anything in common with STEMI outcomes. There are insufficient number of STEMI ECG tracing analyzed to talk about deep learning model, or any machine learning model.

Author Response

Thank you very much for your comment. As you pointed out, we agree that the influence of deep learning model on STEMI reading may be minimal in experienced healthcare professionals who frequently encounter STEMI. We hypothesized that using a deep learning model to read STEMI could be useful in pre-hospital scenarios or for medical personnel who are new with ECG reading. We'll undertake more study by collecting more STEMI ECG numbers, as indicated.

Reviewer 2 Report

The manuscript is interesting to review. The study highlights in patient with STEMI with satisfactory quality (without wandering of base in the ECG) 12 lead ECG recording with deep learning application of AI (Convoluted algorithm) has very high sensitivity and specificity with excellent AUROC. The strength of the study is thorough application and leverage of convoluted deep learning to interpret and diagnose STEMI involving various territories of the coronary arteries. Despite the tenacity of authors, a few concerns remain.

  1. What about positive and negative predictive value? The authors might have included some normal subjects' ECG tracings to calculate it. This will make the manuscript very robust.
  2. What about the patients with baseline ECG abnormalities like LBBB, LVH, AF? As these concomitant ECG changes are not uncommon encounters.

Author Response

Thank you for your detailed and rigorous comments. We considered again for all of your comments.

Point 1:What about positive and negative predictive value? The authors might have included some normal subjects' ECG tracings to calculate it. This will make the manuscript very robust.

Answer:Thank you for the comment. As you mentioned, the positive and negative predictive values were calculated and reported in Table 2. The PPV was only 20.2 % while having a high NPV of more than 99 %. We believe that the low PPV is linked to the cohort's low STEMI prevalence (117 STEMI-ECG vs. 60,040 NSR-ECG; prevalence = 0.2%). The sentences in the methods, results, and discussion sections were all rewritten.

Before revision:

  1. Materials and Methods

2.2. Outcomes

The primary outcome was the diagnostic accuracy of the deep learning model (DLM) for STEMI according to the culprit arteries. The outcomes were measured using area under the receiver operating characteristic curve (AUROC), sensitivity (SEN), and specificity (SPE) using the Youden index.

  1. Results

3.1. Diagnostic accuracy of DLM between STEMI-ECGs versus NSR-ECGs

The DLM also showed a high SEN and SPE, as shown in Table 2. In the detection of the overall STEMI-ECGs, the SEN and SPE were 97.4% and 99.2%, respectively. In the analyses by culprit arteries, all SEN and SPE were in the range of 95-100%.

  1. Discussion

This study demonstrated that DLM could significantly improve the accuracy of STEMI-ECG interpretation. DLM showed high diagnostic performance for STEMI detec-tion (AUROC, 0.998; SEN, 97.4 %; SPE, 99.2 %). In the analysis according to three different culprit coronary arteries, the diagnostic accuracy of LAD, LCX, and RCA STEMI was greater than AUROC 0.99. In addition, the sensitivity and specificity of DLM were > 95 %. We also found that the baseline wander of ECGs could affect the misinterpretation of DLM by analyzing false-positive and false-negative ECGs.

After revision:

  1. Materials and Methods

2.2. Outcomes

The primary outcome was the diagnostic accuracy of the deep learning model (DLM) for STEMI according to the culprit arteries. The outcomes were measured using area under the receiver operating characteristic curve (AUROC), sensitivity (SEN), and specificity (SPE), positive predictive value (PPV), and negative predictive value (NPV) using the Youden index.

  1. Results

3.1. Diagnostic accuracy of DLM between STEMI-ECGs versus NSR-ECGs

The DLM also showed a high SEN and SPE, as shown in Table 2. In the detection of the overall STEMI-ECGs, the SEN and SPE were 97.4% and 99.2%, respectively. In the analyses by culprit arteries, all SEN and SPE were in the range of 95-100%. It demonstrated a very high overall NPV (99.9 %), with the NPV above 99 % regardless of the type of culprit artery. PPV, 

on the other hand, had an extremely low value. (Total: 20.2 %; RCA: 4.6 %; LAD: 15.7 %; LCX: 3.2 %)

  1. Discussion

This study demonstrated that DLM could significantly improve the accuracy of STEMI-ECG interpretation. DLM showed high diagnostic performance for STEMI detec-tion (AUROC, 0.998; SEN, 97.4 %; SPE, 99.2 %). In the analysis according to three different culprit coronary arteries, the diagnostic accuracy of LAD, LCX, and RCA STEMI was greater than AUROC 0.99. In addition, the sensitivity and specificity of DLM were > 95 %. The PPV was only 20.2 % while having a high NPV of more than 99 %. We believe that the low PPV is linked to the cohort's low STEMI prevalence (117 STEMI-ECG vs. 60,040 NSR-ECG; prevalence = 0.2%). We also found that the baseline wander of ECGs could affect the misinterpretation of DLM by analyzing false-positive and false-negative ECGs.

Point 2: What about the patients with baseline ECG abnormalities like LBBB, LVH, AF? As these concomitant ECG changes are not uncommon encounters.

Answer: Thank you for the comment. We agree with your point of view. We used the JACC 4th universal definition of myocardial infarction to enroll in a real STEMI-ECG study. ECGs like LVH and LBBB, according to the definition, can simulate ST deviation. For STEMI-ECG with LBBB, the Sgarbossa criteria has been known. As a result, during the external validation process, we eliminated the STEMI-ECG with LVH or LBBB. The diagnostic accuracy of the deep learning model will be affected if these ECGs are incorporated. Additionally, the proportion of atrial fibrillation in both the STEMI and NSR groups was not discovered in the study, despite the fact that it was not an exclusion criterion. The discussion section been updated to include this constraint.

Before:

  1. Discussion

This study has some limitations. First, the sample size of the STEMI group (n = 117) was smaller than that of the NSR group (n = 60,040) during the same recruitment period. Although the model showed 100 % accuracy in the auto-interpretation of LCX-STEMI, the sample size of LCX-STEMI was smaller than that of LAD-STEMI and RCA-STEMI. There-fore, the accuracy of auto-interpretation of ECG machines or DLM might differ in the study of a larger sample size of STEMI. Second, only one type of ECG machine was used in this study. If the algorithm for auto-interpretation of EGCs varies according to the type or manufacturer of ECG machine, the accuracy of auto-interpretation might change.

After:

  1. Discussion

This study has some limitations. First, the sample size of the STEMI group (n = 117) was smaller than that of the NSR group (n = 60,040) during the same recruitment period. Although the model showed 100 % accuracy in the auto-interpretation of LCX-STEMI, the sample size of LCX-STEMI was smaller than that of LAD-STEMI and RCA-STEMI. Therefore, the accuracy of auto-interpretation of ECG machines or DLM might differ in the study of a larger sample size of STEMI. Second, the control group was selected using the auto-interpretation of ECG machines which resulted in the misinterpretation of 10 cases with STEMI. Third, this was a retrospective single-center study which limits its generali-zability. Fourth, only one type of ECG machine was used in this study. If the algorithm for auto-interpretation of EGCs varies according to the type or manufacturer of ECG machine, the accuracy of auto-interpretation might change. Fifth, Some ECGs with left ventricular hypertrophy (LVH), left bundle branch block (LBBB), and right bundle branch block (RBBB) were not included in the STEMI group. ECGs like LVH and LBBB, according to the myo-cardial infarction definition, can simulate ST deviation [14]. For STEMI-ECG with LBBB, the Sgarbossa criteria has been known. Although the RBBB ECG is not exclusion criteria in the definition of myocardial infarction, we took into account the difficulties of interpreting ST elevation on precordial leads like V1-2 in a clinical setting. As a result, during the ex-ternal validation process, we eliminated the STEMI-ECG with LVH or LBBB. The diagnos-tic accuracy of the deep learning model will be affected if these ECGs are incorporated. Sixth, we used C-statistics to assess the diagnostic accuracy of DLM. Although the C-statistic has been routinely utilized to assess the predictive power of models, its correlation to clinical outcomes has been questioned. Novel statistical indices such as "Net Benefit" have been proposed for measuring the diagnostic performance of tools or the prediction power of models to counteract this flow [21]. To compute "Net Benefit," we attempted to establish a "harm to benefit ratio." We were unable to discover the publication claiming a "harm to benefit ratio" for myocardial infarction despite a comprehensive search. It is recommended that further study be conducted before using a robust prediction model like "Net Benefit."

Reviewer 3 Report

In this study, Choi and colleagues developed a deep learning model (DLM) using more than 130 thousand electrocardiograms and examined it for the prediction of STEMI in 60 thousand electrocardiograms of patients presenting to the emergency room (ER). Their model had excellent diagnostic accuracy with an area under the receiver operating characteristic curve of 0.998. 

The authors should be commended for their authentic effort to address a current clinical challenge in the ER using novel methods like DLM; however, I have the following comments: 

1) Please elaborate and justify the exclusion criteria that are mentioned in Figure 2. Why were LVH or RBBB (exclusion of LBBB makes sense because of different diagnostic criteria of AMI) excluded? Does the developed DLM qualify for detecting STEMI in patients with baseline LVH or RBBB? Please explain "Unable to interpret" and discuss its fairly large portion (54/210). Can it be a limitation of the model?

2) Although the C-statistic has been widely used as a tool for the
evaluation of the prediction power of models, its correlation with clinical outcomes has been questioned. Therefore, novel statistical indices like "Net
Benefit" have been proposed for evaluating the diagnostic performance of methods or the prediction power of models. I suggest keeping the results of
C-statistic as the readers may be more familiar with them and also presenting the results for the net benefit index as it confers a greater clinical weight to the results. This editorial presents a brief comparison between C statistic and net benefit index: https://doi.org/10.1093/eurheartj/ehaa859

3) Page 7, lines 188-194: These lines do not seem relevant to the manuscript or at least, have not been described in the Methods section. Did you run propensity score matching? How it is relevant to false-negative predictions?

4) Please include a brief paragraph about the feasibility and possible implications of this model in daily clinical practice.

5) There are other limitations that should be mentioned: 1) The control group was selected using the auto-interpretation of ECG machines which resulted in the misinterpretation of 10 cases with STEMI. 2) This was a retrospective single-center study which limits its generalizability.

6) The text requires some language editions: 1) Page 2, line 60: Remove spelling and just mention AMI as it has been defined previously; 2) Similarly for NSR-ECG on page 3, line 125; 3) Page 3, line 98: Spell "10" at the beginning of the sentence; etc.

Author Response

Thank you for your detailed and rigorous comments. We considered again for all of your comments.

Point 1:Please elaborate and justify the exclusion criteria that are mentioned in Figure 2. Why were LVH or RBBB (exclusion of LBBB makes sense because of different diagnostic criteria of AMI) excluded? Does the developed DLM qualify for detecting STEMI in patients with baseline LVH or RBBB? Please explain "Unable to interpret" and discuss its fairly large portion (54/210). Can it be a limitation of the model?

Answer: 

Thank you for the comment. We agree with your point of view. We used the JACC 4th universal definition of myocardial infarction to enroll in a real STEMI-ECG study. ECGs like LVH and LBBB, according to the definition, can simulate ST deviation. For STEMI-ECG with LBBB, the Sgarbossa criteria has been known. Although the RBBB ECG is not exclusion criteria in the definition of myocardial infarction, we took into account the difficulties of interpreting ST elevation on precordial leads like V1-2 in a clinical setting. As a result, during the external validation process, we eliminated the STEMI-ECG with LVH, LBBB and RBBB. The diagnostic accuracy of the deep learning model will be affected if these ECGs are incorporated. The discussion section been updated to include this constraint.

"Unable to interpret" in the flow diagram refers to a situation in which there are large baseline wanders. To clarify the meaning, we changed "Unable to interpret" to "Baseline wanders" in Figure 2. 

Discussion before revision:

This study has some limitations. First, the sample size of the STEMI group (n = 117) was smaller than that of the NSR group (n = 60,040) during the same recruitment period. Although the model showed 100 % accuracy in the auto-interpretation of LCX-STEMI, the sample size of LCX-STEMI was smaller than that of LAD-STEMI and RCA-STEMI. There-fore, the accuracy of auto-interpretation of ECG machines or DLM might differ in the study of a larger sample size of STEMI. Second, only one type of ECG machine was used in this study. If the algorithm for auto-interpretation of EGCs varies according to the type or manufacturer of ECG machine, the accuracy of auto-interpretation might change.

Discussion after revision:

This study has some limitations. First, the sample size of the STEMI group (n = 117) was smaller than that of the NSR group (n = 60,040) during the same recruitment period. Although the model showed 100 % accuracy in the auto-interpretation of LCX-STEMI, the sample size of LCX-STEMI was smaller than that of LAD-STEMI and RCA-STEMI. Therefore, the accuracy of auto-interpretation of ECG machines or DLM might differ in the study of a larger sample size of STEMI. Second, the control group was selected using the auto-interpretation of ECG machines which resulted in the misinterpretation of 10 cases with STEMI. Third, this was a retrospective single-center study which limits its generali-zability. Fourth, only one type of ECG machine was used in this study. If the algorithm for auto-interpretation of EGCs varies according to the type or manufacturer of ECG machine, the accuracy of auto-interpretation might change. Fifth, Some ECGs with left ventricular hypertrophy (LVH), left bundle branch block (LBBB), and right bundle branch block (RBBB) were not included in the STEMI group. ECGs like LVH and LBBB, according to the myo-cardial infarction definition, can simulate ST deviation [14]. For STEMI-ECG with LBBB, the Sgarbossa criteria has been known. Although the RBBB ECG is not exclusion criteria in the definition of myocardial infarction, we took into account the difficulties of interpreting ST elevation on precordial leads like V1-2 in a clinical setting. As a result, during the ex-ternal validation process, we eliminated the STEMI-ECG with LVH or LBBB. The diagnos-tic accuracy of the deep learning model will be affected if these ECGs are incorporated. Sixth, we used C-statistics to assess the diagnostic accuracy of DLM. Although the C-statistic has been routinely utilized to assess the predictive power of models, its correlation to clinical outcomes has been questioned. Novel statistical indices such as "Net Benefit" have been proposed for measuring the diagnostic performance of tools or the prediction power of models to counteract this flow [21]. To compute "Net Benefit," we attempted to establish a "harm to benefit ratio." We were unable to discover the publication claiming a "harm to benefit ratio" for myocardial infarction despite a comprehensive search. It is recommended that further study be conducted before using a robust prediction model like "Net Benefit."

Point 2:Although the C-statistic has been widely used as a tool for the
evaluation of the prediction power of models, its correlation with clinical outcomes has been questioned. Therefore, novel statistical indices like "Net
Benefit" have been proposed for evaluating the diagnostic performance of methods or the prediction power of models. I suggest keeping the results of C-statistic as the readers may be more familiar with them and also presenting the results for the net benefit index as it confers a greater clinical weight to the results. This editorial presents a brief comparison between C statistic and net benefit index: https://doi.org/10.1093/eurheartj/ehaa859

Answer:

Thank you for the comment. We agree with your point of view. We attempted to obtain a "harm to benefit ratio" to calculate "Net Benefit" as you advised. Despite a thorough search, we were unable to locate the publication claiming a "harm to benefit ratio" for myocardial infarction. As a result, we expanded the limitation section to include the flaw of C-statistic as well as the value of net benefit. The discussion section been updated to include this constraint. We also add the reference you suggested (reference #21).

Reference

21. Aminorroaya, A.; Tajdini, M.; Masoudkabir, F. Time for clinicians to revisit their perspectives on C-statistic. Eu-ropean Heart Journal 2020, 42, 132-133, doi:10.1093/eurheartj/ehaa859.

Discussion before:

This study has some limitations. First, the sample size of the STEMI group (n = 117) was smaller than that of the NSR group (n = 60,040) during the same recruitment period. Although the model showed 100 % accuracy in the auto-interpretation of LCX-STEMI, the sample size of LCX-STEMI was smaller than that of LAD-STEMI and RCA-STEMI. There-fore, the accuracy of auto-interpretation of ECG machines or DLM might differ in the study of a larger sample size of STEMI. Second, only one type of ECG machine was used in this study. If the algorithm for auto-interpretation of EGCs varies according to the type or manufacturer of ECG machine, the accuracy of auto-interpretation might change.

Discussion after:

This study has some limitations. First, the sample size of the STEMI group (n = 117) was smaller than that of the NSR group (n = 60,040) during the same recruitment period. Although the model showed 100 % accuracy in the auto-interpretation of LCX-STEMI, the sample size of LCX-STEMI was smaller than that of LAD-STEMI and RCA-STEMI. Therefore, the accuracy of auto-interpretation of ECG machines or DLM might differ in the study of a larger sample size of STEMI. Second, the control group was selected using the auto-interpretation of ECG machines which resulted in the misinterpretation of 10 cases with STEMI. Third, this was a retrospective single-center study which limits its generali-zability. Fourth, only one type of ECG machine was used in this study. If the algorithm for auto-interpretation of EGCs varies according to the type or manufacturer of ECG machine, the accuracy of auto-interpretation might change. Fifth, Some ECGs with left ventricular hypertrophy (LVH), left bundle branch block (LBBB), and right bundle branch block (RBBB) were not included in the STEMI group. ECGs like LVH and LBBB, according to the myo-cardial infarction definition, can simulate ST deviation [14]. For STEMI-ECG with LBBB, the Sgarbossa criteria has been known. Although the RBBB ECG is not exclusion criteria in the definition of myocardial infarction, we took into account the difficulties of interpreting ST elevation on precordial leads like V1-2 in a clinical setting. As a result, during the ex-ternal validation process, we eliminated the STEMI-ECG with LVH or LBBB. The diagnos-tic accuracy of the deep learning model will be affected if these ECGs are incorporated. Sixth, we used C-statistics to assess the diagnostic accuracy of DLM. Although the C-statistic has been routinely utilized to assess the predictive power of models, its correlation to clinical outcomes has been questioned. Novel statistical indices such as "Net Benefit" have been proposed for measuring the diagnostic performance of tools or the prediction power of models to counteract this flow [21]. To compute "Net Benefit," we attempted to establish a "harm to benefit ratio." We were unable to discover the publication claiming a "harm to benefit ratio" for myocardial infarction despite a comprehensive search. It is recommended that further study be conducted before using a robust prediction model like "Net Benefit."

Point 3: Page 7, lines 188-194: These lines do not seem relevant to the manuscript or at least, have not been described in the Methods section. Did you run propensity score matching? How it is relevant to false-negative predictions?

Answer:Thank you for your feedback. This is an error, and the sentences have no bearing on our paper. As a result, the sentences in the Methods section were deleted.

Before:

  1. Results

3.2. analysis for false-positive and false-negative results

Three STEMI-ECGs were false-negative by DLM. Two ECGs were the inferior wall (all RCA culprit arteries) and an anteroseptal wall STEMI (LAD culprit artery). All false-negative ECG results showed baseline wanders. For the mechanical CPR with LU-CASTM and manual CPR comparison, 1:1 PSM was applied to the two imbalanced covari-ates (bystander CPR and ECMO). Both groups included 149 patients. There were no im-balanced covariates in either group after PSM. In terms of outcomes, the LUCASTM group showed a lower rate of sustained ROSC than the manual CPR (27.5 vs. 46.3%, p = 0.001); however, there was no significant difference across groups for survival at discharge (2.7 vs. 5.4%, p = 0.377).

After revision:

  1. Results

3.2. analysis for false-positive and false-negative results

Three STEMI-ECGs were false-negative by DLM. Two ECGs were the inferior wall (all RCA culprit arteries) and an anteroseptal wall STEMI (LAD culprit artery). All false-negative ECG results showed baseline wanders. For the mechanical CPR with LU-CASTM and manual CPR comparison, 1:1 PSM was applied to the two imbalanced covari-ates (bystander CPR and ECMO). Both groups included 149 patients. There were no im-balanced covariates in either group after PSM. In terms of outcomes, the LUCASTM group showed a lower rate of sustained ROSC than the manual CPR (27.5 vs. 46.3%, p = 0.001); however, there was no significant difference across groups for survival at discharge (2.7 vs. 5.4%, p = 0.377).

Point 4:Please include a brief paragraph about the feasibility and possible implications of this model in daily clinical practice

Answer:Thank you for the comment. The deep learning model allows the possibility of STEMI to be detected with high accuracy in a short period in circumstances when it is difficult to determine the STEMI on the ECG in the pre-hospital stage, or for medical professionals who are unfamiliar with the ECG reading at the primary medical institution. This might help STEMI patients by allowing them to go to a cardiovascular center where revascularization can be performed quickly. We added a brief paragraph about the feasibility and possible implications of this model in daily clinical practice in the Discussion section. 

Before:

  1. Discussion

The analysis of false-positive cases also showed that the control group included 10 STEMI (3 ECGs without baseline wander and 7 ECGs with baseline wander). This result was caused by setting the control group using the auto-interpretation of ECG machines. In addition to the 10 STEMI ECGs, true-negative results included 402 NSR-ECGs. Unfortu-nately, we could not identify the cause of false-positive results because the DLM did not report the analysis of misinterpretation. Additionally, the ST-elevation of ECGs without baseline wander was significantly higher than that of ECGs with baseline wander. This result also demonstrated that the baseline wander could significantly interfere with the interpretation of ST-elevation by DLM [20].

After:

  1. Discussion

The analysis of false-positive cases also showed that the control group included 10 STEMI (3 ECGs without baseline wander and 7 ECGs with baseline wander). This result was caused by setting the control group using the auto-interpretation of ECG machines. In addition to the 10 STEMI ECGs, true-negative results included 402 NSR-ECGs. Unfortu-nately, we could not identify the cause of false-positive results because the DLM did not report the analysis of misinterpretation. Additionally, the ST-elevation of ECGs without baseline wander was significantly higher than that of ECGs with baseline wander. This result also demonstrated that the baseline wander could significantly interfere with the interpretation of ST-elevation by DLM [20].

The DLM allows the possibility of STEMI to be detected with high accuracy in a short period in circumstances when it is difficult to determine the STEMI on the ECG in the pre-hospital stage, or for medical professionals who are unfamiliar with the ECG reading at the primary medical institution. This might help STEMI patients by allowing them to go to a cardiovascular center where revascularization can be performed quickly.

Point 5: There are other limitations that should be mentioned: 1) The control group was selected using the auto-interpretation of ECG machines which resulted in the misinterpretation of 10 cases with STEMI. 2) This was a retrospective single-center study which limits its generalizability.

Answer:Thank you for the comment. Other limitations you suggested are listed to the Discussion section. 

Before:This study has some limitations. First, the sample size of the STEMI group (n = 117) was smaller than that of the NSR group (n = 60,040) during the same recruitment period. Although the model showed 100 % accuracy in the auto-interpretation of LCX-STEMI, the sample size of LCX-STEMI was smaller than that of LAD-STEMI and RCA-STEMI. There-fore, the accuracy of auto-interpretation of ECG machines or DLM might differ in the study of a larger sample size of STEMI. Second, only one type of ECG machine was used in this study. If the algorithm for auto-interpretation of EGCs varies according to the type or manufacturer of ECG machine, the accuracy of auto-interpretation might change.

After:

This study has some limitations. First, the sample size of the STEMI group (n = 117) was smaller than that of the NSR group (n = 60,040) during the same recruitment period. Although the model showed 100 % accuracy in the auto-interpretation of LCX-STEMI, the sample size of LCX-STEMI was smaller than that of LAD-STEMI and RCA-STEMI. Therefore, the accuracy of auto-interpretation of ECG machines or DLM might differ in the study of a larger sample size of STEMI. Second, the control group was selected using the auto-interpretation of ECG machines which resulted in the misinterpretation of 10 cases with STEMI. Third, this was a retrospective single-center study which limits its generali-zability. Fourth, only one type of ECG machine was used in this study. If the algorithm for auto-interpretation of EGCs varies according to the type or manufacturer of ECG machine, the accuracy of auto-interpretation might change. Fifth, Some ECGs with left ventricular hypertrophy (LVH), left bundle branch block (LBBB), and right bundle branch block (RBBB) were not included in the STEMI group. ECGs like LVH and LBBB, according to the myo-cardial infarction definition, can simulate ST deviation [14]. For STEMI-ECG with LBBB, the Sgarbossa criteria has been known. Although the RBBB ECG is not exclusion criteria in the definition of myocardial infarction, we took into account the difficulties of interpreting ST elevation on precordial leads like V1-2 in a clinical setting. As a result, during the ex-ternal validation process, we eliminated the STEMI-ECG with LVH or LBBB. The diagnos-tic accuracy of the deep learning model will be affected if these ECGs are incorporated. Sixth, we used C-statistics to assess the diagnostic accuracy of DLM. Although the C-statistic has been routinely utilized to assess the predictive power of models, its correlation to clinical outcomes has been questioned. Novel statistical indices such as "Net Benefit" have been proposed for measuring the diagnostic performance of tools or the prediction power of models to counteract this flow [21]. To compute "Net Benefit," we attempted to establish a "harm to benefit ratio." We were unable to discover the publication claiming a "harm to benefit ratio" for myocardial infarction despite a comprehensive search. It is recommended that further study be conducted before using a robust prediction model like "Net Benefit."

Point 6: The text requires some language editions: 1) Page 2, line 60: Remove spelling and just mention AMI as it has been defined previously; 2) Similarly for NSR-ECG on page 3, line 125; 3) Page 3, line 98: Spell "10" at the beginning of the sentence; etc.

Answer:Thank you for the comment. As you asked, we corrected the spelling errors

Before:

  • Page 2, line 60

: By searching electronic medical records during the recruitment period, we recruited patients with acute myocardial infarction (AMI) who received more than one stent insertion for culprit artery occlusion after visiting the emergency room in the Hallym University Kangnam Sacred Heart Hospital.

  • Page 3, line 125

: A total of 60,157 ECGs were obtained. These included 117 STEMI-ECGs and 60,040 normal sinus rhythm (NSR)-ECGs

  • Page 3, line 98

: 10 seconds ECG was split to obtain four ensembled outputs for 2.5 seconds each to ensure stable performance.

After revision:

  • Page 2, line 60

: By searching electronic medical records during the recruitment period, we recruited patients with AMI who received more than one stent insertion for culprit artery occlusion after visiting the emergency room in the Hallym University Kangnam Sacred Heart Hospital.

  • Page 3, line 125

: A total of 60,157 ECGs were obtained. These included 117 STEMI-ECGs and 60,040 NSR-ECGs

  • Page 3, line 98

: Ten seconds ECG was split to obtain four ensembled outputs for 2.5 seconds each to ensure stable performance.

Round 2

Reviewer 2 Report

The manuscript is improved a lot, I am sure the manuscript once published will inform the guidelines. I am happy to review the manuscript. I congratulate you for your outstanding work.

Reviewer 3 Report

Thanks for the detailed and delicate revision. No additional comments.